# Structured follow-up pathway to address unmet needs after transient ischaemic attack and minor stroke (SUPPORT TIA): Feasibility study and process evaluation

Grace M. Turner[1,2]*, Melanie Calvert[3,2,4,5,6], Robbie Foy[7], Lou Atkins[8], Philip Collis[9], Sarah Tearne[10], Sue Jowett[11], Kelly Handley[10], Jonathan Mant[12]

**1** School of Sport, Exercise and Rehabilitation Science, University of Birmingham, Birmingham, United Kingdom, **2** Centre for Patient Reported Outcomes Research, University of Birmingham, Birmingham, United Kingdom, **3** School of Health Sciences, University of Birmingham, Birmingham, United Kingdom, **4** NIHR Applied Research Collaboration West Midlands, Birmingham, United Kingdom, **5** NIHR Birmingham Biomedical Research Centre, University Hospitals Birmingham NHS Foundation Trust and University of Birmingham, Birmingham, United Kingdom, **6** Birmingham Health Partners Centre for Regulatory Science and Innovation, University of Birmingham, Birmingham, United Kingdom, **7** Leeds Institute of Health Sciences, University of Leeds, Leeds, United Kingdom, **8** Centre for Behaviour Change, University College London, London, United Kingdom, **9** Patient partner, United Kingdom, **10** Birmingham Clinical Trials Unit, University of Birmingham, Birmingham, United Kingdom, **11** Health Economics Unit, University of Birmingham, Birmingham, United Kingdom, **12** Department of Public Health and Primary Care, University of Cambridge, Cambridge, United Kingdom

* g.turner.1@bham.ac.uk

## Abstract

### Background

Care following transient ischaemic attack (TIA) and minor stroke is variable and often leaves patients feeling abandoned and uncertain. We developed a theoretically-informed, multifaceted intervention which comprised nurse-led, structured follow-up at 4 weeks after TIA/minor stroke to identify and address patient needs. This study evaluated the feasibility and acceptability of both the intervention and procedures to inform a future randomised controlled trial.

### Method

We conducted a multicentre, randomised feasibility study with mixed-methods process evaluation (ISRCTN registry reference: ISRCTN39864003). We collected patient reported outcome measures (PROMs) at 1, 12 and 24 weeks and clinical data at baseline and 24 weeks. The process evaluation comprised qualitative interviews with a sub-sample, feed-back questionnaires, and observations of intervention delivery.

### Results

We recruited 54 patients over 12 months, achieving 90% of the target sample size (n = 60). PROMs return rates were 94.4% (51/54), 85.2% (46/54) and 71.1% (27/38) at 1, 12, and 24-weeks, respectively. Intervention fidelity was high and the intervention largely aligned

**Data availability statement:** All relevant data are within the manuscript and its Supporting Information files.

**Funding:** GT received funding from the National Institute for Health and Care Research (NIHR) Post-Doctoral Fellowship Scheme grant number PDF-2017-10-047. https://www.nihr.ac.uk/ The funders had no role in study design, data collection and analysis, decision to publish, or preparation of the manuscript.

**Competing interests:** GT received grant funding to deliver this project from a National Institute for Health and Care Research (NIHR) Post-Doctoral Fellowship Scheme grant number PDF-2017-10-047. RF receives grant funding from the NIHR (with funds paid to his institution) and chairs the NICE Implementation Strategy Group (non-paid). SJ reports grants from NIHR and Wellcome outside the submitted work. MC receives grant funding from the NIHR Birmingham Biomedical Research Centre, NIHR Applied Research Collaboration West Midlands, NIHR BTRU Precision Transplant and Cellular Therapeutics, Health Data Research UK, Innovate UK, Macmillan Cancer Support, GSK, UCB Pharma, Research England, European Commission and EFPIA, Brain Tumor Charity, Gilead, Janssen, NIHR, UKRI, UK Research and Innovation, Merck; Royalties or licenses from Symptom Burden Questionnaire-Long COVID (as part of development team received revenue share from commercial license); Consulting fees from Aparito Ltd, Boehringer Ingelheim, CIS Oncology, Takeda, Merck, Daiichi Sankyo, Glaukos, GSK, PCORI, Genentech, Vertex, ICON, Halfloop, Pfizer; Payment or honoraria for lecture fees from University of Maastricht, reviewer fees from South-Eastern Norway Regional Health Authority and Singapore National Medical Research Council, speaker fee from Cochrane Portugal. PC, JM, KH and LA declare no conflict of interest.

with the theoretical underpinnings. The process evaluation illustrated how patients benefitted from the intervention through support they would not have received through usual care. This included direct referral or signposting to support services, information and education, actionable advice, and reassurance about and normalisation of recovery. The trial design was feasible and acceptable for both patients and clinicians.

## Conclusion

Nurse-led, structured follow-up after TIA and minor stroke is feasible, acceptable and valued by patients and clinicians. Our intervention can identify and help address unmet needs. A definitive randomised trial to evaluate intervention effectiveness and cost-effectiveness is feasible and acceptable.

## Introduction

Transient ischaemic attack (TIA) and minor stroke are important risk factors for stroke. Over 46,000 people experience a first TIA or minor stroke per year in the United Kingdom (UK), [1] 240,000 in the United States [2] and 310,000 in China [3].

National guidelines, including those from Europe and the United States, promote long-term management that focuses on stroke prevention. [4–6] However, people affected by TIA and minor stroke feel unsupported in managing their medication and lifestyle changes for stroke prevention, and often lack basic understanding of their diagnosis, stroke risk and preventative medication. [7] Qualitative research indicates information is difficult to process at the time of diagnosis; language was too medical; clinicians gave contradictory advice; and information was too generic and not personalised.(7) Furthermore, many experience a variety of residual impairments and unmet needs, including anxiety, low mood, fatigue, cognitive impairment, physical weakness, visual impairment and impaired speech. [8–16] These impairments can affect people's work return and performance, social activities and relationships with friends and family. [8,13–19] This is further exacerbated by, loss of confidence, changes in family roles and dynamics and driving restrictions.(7) After TIA and minor stroke people feel abandoned following hospital discharge with variable follow-up care. [7] Clinicians commonly lack awareness of residual problems and holistic needs after TIA and minor stroke.(7)

Care needs after TIA and minor stroke include information provision (diagnosis and stroke risk); stroke prevention (medication and lifestyle change); and holistic care (residual problems and return to work or usual activities). However, it is uncertain how to best support these patients after rapid specialist review in hospital. We therefore developed a multifaceted intervention to identify and address unmet needs after TIA and minor stroke: SUPPORT TIA (Structured follow-Up Pathway to imProve management Of Residual impairmenTs and patients' quality of life after Transient Ischaemic Attack and minor stroke).

In accordance with UK Medical Research Council (MRC) guidance on developing and evaluating complex interventions, [20] we evaluated the feasibility and acceptability of both the intervention and procedures for a future randomised controlled trial (RCT). We also evaluated intervention fidelity and contextual influences on delivery.

## Methods

We conducted a multicentre, individual randomised feasibility study with a mixed-methods process evaluation. Objectives are shown in Table 1. Favourable ethical opinion was gained from the Wales Research Ethics Committee (REC) 1 (23/02/2021, REC reference: 21/

**Table 1. Feasibility and process evaluation objectives, outcomes and measurement of outcomes.**

| Objective | Feasibility outcomes | Measurement of outcome |
|---|---|---|
| **Feasibility study** | | |
| a) Assess feasibility and acceptability of the trial methods | Number of eligible/ineligible patients and reasons for ineligibility | Recruitment log |
| | Proportion of participants who consent face-to-face, verbal or postal | Registration log: method of consent |
| | Willingness of clinical staff to randomise patients | Interviews (clinical staff involved in randomisation) |
| | Recruitment and attrition rates | Registration log |
| | Response rates and frequencies of missing data: participant completed questionnaires and case report forms | 1, 12 and 24 week questionnaires Case Report forms |
| | End of study clinic appointment attendance rates | End of Study Clinic Appointment Form |
| | Acceptability of the trial design | Interviews (participants and clinical staff) Structured observations |
| b) Inform the sample size for a definitive trial | Standard deviations of continuous patient reported outcome measures at six months | Patient reported outcome measure scores |
| | Recruitment and attrition rates | Registration log |
| c) Inform selection of the primary outcome measure for a definitive randomised controlled trial | Correlation of patient reported outcome measures | Patient reported outcome measure scores |
| | Patient reported outcome measure response rates and missing data | 1, 12 and 24 week questionnaires |
| **Process evaluation** | | |
| d) Investigate intervention acceptability to participants and intervention providers | Participants' and intervention providers' opinion on acceptability of the intervention | Interviews (participants and intervention providers) Feedback questionnaire (intervention participants) |
| e) Test hypotheses relating to the theoretical underpinning of the intervention | Participants' satisfaction with identification and management of needs | Interviews (participants and intervention providers) Feedback questionnaire (intervention participants) |
| | Participants acting on agreed action plans and/or accessing support services | Interviews (participants) |
| f) Assess adequacy of training for intervention providers | Intervention providers' understanding of the intervention components | Interviews (intervention providers) |
| g) Assess adherence to the intervention | Intervention providers' adherence to and deviations from the intervention manual | Structured observations Intervention log |
| h) Assess contamination with the control group | Control group contamination | Interviews (participants and clinical staff) Structured observations |
| i) Define intervention dose" | Intervention follow-up appointment: attendance, length of appointment and number of appointments | Intervention log |
| j) Explore participant receipt and understanding of the intervention | Participants' perception of the intervention | Interviews (participants) Feedback questionnaire (intervention participants) |
| k) Explore to what extent the intervention was enacted as intended by intervention participants | Participants acting on agreed action plans and/or accessing support services | Interviews (intervention participants) |

WA/0036). The trial was registered on the ISRCTN registry (Reference: ISRCTN39864003). We summarise methods from the published protocol [21].

## Participants

Participants were recruited between 1st September 2021 and 31st August 2022 from TIA clinics and stroke wards at one hospital in South East England (Berkshire) and two in North West England (Wigan and Liverpool). Participants were adults who had experienced a first or recurrent TIA or minor stroke, had ability to converse in everyday English and read in English and capacity to provide fully informed consent. Full eligibility criteria are reported elsewhere. [21]

Patient medical records were screened for eligibility and potential participants were invited to take part in the study face-to-face or by phone. Informed consent was taken face-to-face (for people approached in clinic), by post (for people needing more time to consider participation) or verbally (for people approached via phone).

## Intervention

The intervention, detailed elsewhere, [21] comprised six main components summarised in the logic model (Fig 1). Intervention development was underpinned by the Behaviour Change Wheel theoretical framework [22] and iteratively refined in collaboration with patient partners and a multidisciplinary team.

| Intervention component | Mechanism of change | Outputs | Potential outcomes |
|---|---|---|---|
| 1. Training for nurses or AHPs delivering the intervention. | HCPs educated about potential needs after TIA or minor stroke, including health and social consequences. HCPs instructed how to identify and address potential unmet needs, including use of intervention materials. | ↑ HCPs knowledge of potential needs and strategies to identify and address needs. | *Short term* <br> • ↑ satisfaction with care. <br> • Access relevant further support (e.g. GP, support service). <br> • Access relevant resources (e.g. websites, apps). <br> • ↑ knowledge and understanding of diagnosis and stroke risk. |
| 2. Structured nurse or AHP led follow-up appointment, 4 weeks after TIA or minor stroke. | Service to provide patients with access to nurse or AHP follow-up. | Patients access holistic care and support for needs related to their TIA or minor stroke. Needs may be actioned immediately (e.g. driving information, reassurance) or an agreed plan for self-management or further support. | |
| 3. Needs checklist completed by participants prior to the appointment. | Checklist provides patients with the opportunity to reflect on their needs and facilitates communication with the HCP. | Patients' needs are actively identified and acknowledged by the HCP. | *Medium term* <br> • ↑ medication adherence. <br> • ↑ confidence and ability to self-manage needs. |
| 4. Resources to support management of needs, including a website of resources and support services; list of local support services; and a self-management booklet. | ↑ knowledge of relevant support services and resources. | Patients are referred or signposted to relevant support services. Patients are recommended relevant resources for their individual needs (e.g. websites, apps). | *Long term* <br> • ↑ health related quality of life. <br> • ↓ or improved residual problems (e.g. anxiety, fatigue). <br> • ↓ in stroke risk factors. |
| 5. Action plan. | Provides an opportunity for shared decision making and goal setting, and empowers patient to access services or resources and/or self-manage. | Patients are given clear actions to self-manage needs or access further support. | |
| 6. Structured GP letter | ↑ communication between secondary and primary care, and patients and GPs. Patients are empowered to access GP support. | GPs have better understanding of their patients' needs and care received. GPs receive clear recommendations for how to further support patient needs. | |

AHP: Allied Health Professional; GP: General Practitioner; HCP: Healthcare provider; TIA: Transient Ischaemic Attack; ↑: increase; ↓: decrease

**Fig 1. Logic model.**

## Control

Participants randomised to control received usual care and were posted a Stroke Association TIA information sheet.

## Randomisation

Participants were randomised 1:1 to the intervention or control arm. Randomisation was provided by telephone or by a secure online randomisation system at the Birmingham Clinical Trials Unit. To ensure that these potential confounding factors were equally distributed between groups, a minimisation algorithm was used with the following variables: age at consent (<60 years, ≥60 years); sex (male, female); diagnosis (TIA, minor stroke); employment (employed, non-employed/retired). A 'random element' was included in the minimisation algorithm.

Participants were randomised at baseline by clinical staff; however, to prevent baseline patient reported outcomes being affected by study arm allocation, participants were notified of their randomisation allocation after they returned the 1-week questionnaire or after 3 weeks (if the 1-week questionnaire was not returned). For the duration of the study, interim data were reviewed, in strict confidence, by an independent Study Oversight Committee.

## Outcomes and data collection

**Feasibility outcomes and process evaluation.** The feasibility study and process evaluation outcomes are detailed in Table 1, the latter based on the National Institutes of Health (NIH) Behavioural Change Consortium (BCC) treatment fidelity framework [23] and including intervention design, training, delivery, receipt and enactment.

**Structured observations.** A member of the study team (GT) observed a sample of intervention appointments remotely via teleconference. A checklist was used to document adherence to the protocol. It was originally planned that a study team member would also observe recruitment and consent procedures and end of study clinic appointments; however, due to COVID restrictions this was not possible. Instead, these study processes were discussed in qualitative interviews with clinical staff.

**Intervention feedback questionnaires.** Intervention feedback questionnaires were sent to participants after the intervention. These included 5-point Likert scale questions (e.g., strongly agree – strongly disagree) and free text questions about experiences of the checklist, appointment and action plan.

**Qualitative interviews.** Semi-structured interviews were conducted with eight intervention participants, three control participants and four clinical staff to explore acceptability of the intervention and trial design. Interviews were conducted by GT, an experienced qualitative researcher, remotely by telephone or video call. Five people who declined to participate were asked to report their reasons for declining.

## Baseline, patient reported, health economic and clinical outcome measures

Table 2 summarises the baseline, patient reported, health economic and clinical outcome measures. Follow-up questionnaires were completed by participants, either by post or electronically, at 1, 12 and 24 weeks.

## Sample size

We aimed to recruit 60 participants (30 intervention arm, 30 control arm). As this was a feasibility study, no formal sample size calculation was performed; however, the sample size was

the estimated number that would demonstrate the potential for sufficient recruitment rate of our target population. [24]

## Analysis

Quantitative data were analysed using simple descriptive statistics. Resources required to deliver the intervention were estimated. Correlation of standard deviations (SD) for the following patient reported outcome measures were calculated: Patient-Reported Outcomes Measurement Information System (PROMIS) Mental Health; PROMIS Physical Health; EuroQol 5-Dimensions (EQ-5D-5L); Hospital Anxiety and Depression Scale (HADS); Fatigue Assessment Scale (FAS). Analyses were conducted using Stata v18.

For qualitative data, interviews were audio recorded and transcribed verbatim. NVivo v12 was used to manage, sort, code and organise the anonymised transcribed data. Interview transcripts were analysed by GT using directed thematic analysis, using Braun and Clarke's 6-stage process,[25] informed by the research aims.[26]

Quantitative and qualitative data were initially analysed in isolation. On completion of all analyses, the data were then brought together; qualitative data were used to provide further detail and highlight possible explanations for the quantitative findings.

**Table 2. Summary of baseline, patient reported, health economic and clinical outcome measures.**

| | Data | Timepoint |
|---|---|---|
| **Baseline data** | Contact details | Baseline (from medical records or participant interview, by a member of clinical staff) |
| | Demographic: date of birth, sex, ethnicity, employment status | |
| | Medical: diagnosis, date of TIA or minor stroke, modified Rankin scale score, length of stay, smoking status, alcohol consumption, height, weight, body mass index, comorbidities, medication, blood pressure, cholesterol | |
| **Patient reported outcome measure** | Patient-Reported Outcomes Measurement Information System (PROMIS)- Global Health 10 | 1, 12 and 24 weeks (completed by participants, either by post or electronically) |
| | EuroQol 5-Dimensions (EQ-5D-5L) | |
| | Hospital Anxiety and Depression Scale (HADS) | |
| | Fatigue Assessment Scale (FAS) | |
| | Patient Activation Measure-13 (PAM-13) | |
| | Medication Adherence Rating Scale -5 (MARS) | |
| | Satisfaction with overall care after TIA/minor stroke question: 5-point Likert scale (very satisfied – very dissatisfied) | |
| **Health economics** | Use of healthcare services | 12 and 24 weeks (completed by participants, either by post or electronically) |
| | Change in employment status, altered work hours and days off sick | |
| | Other costs incurred because of TIA or minor stroke | |
| **Clinical data** | Body Mass Index | Baseline and 26 weeks (by a research nurse or clinical staff) |
| | Blood pressure | |
| | Bloods: cholesterol | |
| | Medications | |

### Progression criteria

Pre-defined progression criteria were agreed by the Study Oversight Committee and followed a traffic light system using quantitative measures supported by qualitative data (S2: Table S1).

### Patient and public involvement

A core group of three people who have experienced TIA or minor stroke supported this study from inception, with *ad hoc* contributions from a wider group of people with experience of TIA or minor stroke. The group supported the initial development of the research question and funding application, informed by their priorities and experiences. The group was involved in: selection of outcome measures; development of study documents; and design of the trial, such as recruitment strategies and considering participant burden related to data collection and attending intervention appointments. The group was integral to the intervention development, in particular the website of support services and resources. The group supported the delivery of the study, in particularly contributing to discussions about slow recruitment. One member (PC) is a co-author and member of the Study Oversight Committee.

## Results

### Trial design: feasibility and acceptability

**Recruitment.** Overall, 823 patients were screened for eligibility and 54 participants were recruited over 12 months, achieving 90% of the target sample size (n = 60). A sub-sample of 11 participants and 4 site staff participated in qualitative interviews (S2: Tables S2-3)

Number of eligible vs ineligible patients and reasons for ineligibility were reported by two sites (S2: Table S4). From these two sites, 694 patients were screened; of these 177 met the eligibility criteria (25.5%; 177/694) and 40 were recruited (22.6% 40/177). The main reasons for ineligibility were no confirmed diagnosis of TIA or minor stroke and previous history of stroke.

Of the 54 participants recruited, the average age was 70.2 (SD 11.2) years, 44.4% (24/54) were male and all were white ethnicity. Three quarters of participants had a diagnosis of TIA (75.9%; 41/54). Full demographic and characteristics are reported in Table 3.

The median recruitment per month was 5 (Interquartile Range [IQR] 5, 6; range 4-10) (S2: Table S5). Most participants consented by post 63.0% (33/54) (face-to-face 35.2%; 19/54, verbal 2.0%; 1/54) (S2: Table S6). Participants' main motivations to take part were to help other people and to access additional care.

> *"Well, because I thought it would be of benefit to people and staff, in the future. And I thought it would be of benefit to me, as well."* [P8, intervention]

Participants were satisfied with both telephone and face-to-face recruitment process and felt well informed, *"She explained it all very thoroughly"* [P10, control]. One participant described being approached to participate in the study as "*the best thing that's happened to me in that whole 36 hours* [on the ward]" [P6, intervention].

Clinical staff reported recruitment processes as *"straightforward"* [N4]. Barriers to recruitment were: being unable to contact patients after discharge, delays in consultants confirming the diagnosis, patients having work or caring commitments, and patients who "*just didn't believe the diagnosis*" [N2]. This was corroborated by interviews with five people who declined to participate.

**Table 3. Baseline demographic and clinical characteristics (n = 54).**

| | | Intervention (N = 25) | Control (N = 29) | Overall (N = 54) |
|---|---|---|---|---|
| **Demographic information** | | | | |
| Age, years | Mean (SD) | 73.0 (12.1) | 67.7 (9.9) | 70.2 (11.2) |
| Sex | Male | 10 (40.0) | 14 (48.3) | 24 (44.4) |
| | Female | 15 (60.0) | 15 (51.7) | 30 (55.6) |
| Ethnic Group | White | 25 (100.0) | 29 (100.0) | 54 (100.0) |
| Employment status | Employed- full time | 5 (20.0) | 7 (24.1) | 12 (22.2) |
| | Employed- part time | 1 (4.0) | 4 (13.8) | 5 (9.3) |
| | Retired | 17 (68.0) | 17 (58.6) | 34 (63.0) |
| | Unemployed | 1 (4.0) | 1 (3.4) | 2 (3.7) |
| | Other | 1 (4.0) | 0 (0.0) | 1 (1.9) |
| **TIA/ minor stroke information** | | | | |
| Diagnosis | TIA | 19 (76.0) | 22 (75.9) | 41 (75.9) |
| | Minor stroke | 6 (24.0) | 7 (24.1) | 13 (24.1) |
| Modified Rankin scale score | 0 | 14 (56.0) | 18 (62.1) | 32 (59.3) |
| | 1 | 7 (28.0) | 8 (27.6) | 15 (27.8) |
| | 2 | 3 (12.0) | 2 (6.9) | 5 (9.3) |
| | 3 | 1 (4.0) | 1 (3.4) | 2 (3.7) |
| Hospital length of stay | <24 hours | 19 (76.0) | 18 (62.1) | 37 (68.5) |
| | 1 night | 0 (0.0) | 3 (10.3) | 3 (5.6) |
| | 2 nights | 1 (4.0) | 2 (6.9) | 3 (5.6) |
| | 3-4 nights | 1 (4.0) | 0 (0.0) | 1 (1.9) |
| | 5-6 nights | 0 (0.0) | 1 (3.4) | 1 (1.9) |
| | ≥7 nights | 0 (0.0) | 1 (3.4) | 1 (1.9) |
| | Missing | 4 (16.0) | 4 (13.8) | 8 (14.8) |
| **Risk factors and medical history** | | | | |
| Smoking status | Smoker | 5 (20.0) | 5 (17.2) | 10 (18.5) |
| | Non-smoker | 14 (56.0) | 13 (44.8) | 27 (50.0) |
| | Ex-smoker | 6 (24.0) | 10 (34.5) | 16 (29.6) |
| | Missing | 0 (0.0) | 1 (3.4) | 1 (1.9) |
| Alcohol consumption, units per week | 0 | 11 (44.0) | 13 (44.8) | 24 (44.4) |
| | 1-2 | 2 (8.0) | 5 (17.2) | 7 (13.0) |
| | 3-4 | 6 (24.0) | 4 (13.8) | 10 (18.5) |
| | 5-6 | 2 (8.0) | 2 (6.9) | 4 (7.4) |
| | 7-9 | 0 (0.0) | 3 (10.3) | 3 (5.6) |
| | ≥10 | 4 (16.0) | 1 (3.4) | 5 (9.3) |
| | Missing | 0 (0.0) | 1 (3.4) | 1 (1.9) |
| BMI, kg/m2 | Healthy (18.5-25.9) | 11 (44.0) | 7 (24.1) | 18 (33.3) |
| | Underweight (<18.5) | 0 (0.0) | 1 (3.4) | 1 (1.9) |
| | Overweight (26-30) | 4 (16.0) | 11 (37.9) | 15 (27.8) |
| | Obese (>30) | 7 (28.0) | 8 (27.6) | 15 (27.8) |
| | Missing | 3 (12.0) | 2 (6.9) | 6 (11.1) |
| Atrial fibrillation | Yes | 23 (92.0) | 25 (86.2) | 48 (88.9) |
| CKD | Yes | 1 (4.0) | 0 (0.0) | 1 (1.9) |
| COPD | Yes | 2 (8.0) | 2 (6.9) | 4 (7.4) |
| CHD | Yes | 3 (12.0) | 4 (13.8) | 7 (13.0) |
| Depression | Yes | 4 (16.0) | 3 (10.3) | 7 (13.0) |
| Anxiety | Yes | 5 (20.0) | 5 (17.2) | 10 (18.5) |

*(Continued)*

**Table 3.** (Continued)

| | | Intervention (N = 25) | Control (N = 29) | Overall (N = 54) |
|---|---|---|---|---|
| Diabetes mellitus Type 2 | Yes | 4 (16.0) | 6 (20.7) | 10 (18.5) |
| Epilepsy | Yes | 0 (0.0) | 1 (3.4) | 1 (1.9) |
| Heart failure | Yes | 0 (0.0) | 1 (3.4) | 1 (1.9) |
| Hypertension | Yes | 13 (52.0) | 16 (55.2) | 29 (53.7) |
| Hyperthyroidism | Yes | 2 (8.0) | 3 (10.3) | 5 (9.3) |
| Osteoporosis | Yes | 1 (4.0) | 0 (0.0) | 1 (1.9) |
| PAD | Yes | 0 (0.0) | 1 (3.4) | 1 (1.9) |
| Rheumatoid arthritis | Yes | 0 (0.0) | 0 (0.0) | 0 (0.0) |
| Blood pressure: systolic (mmHg) | Median [IQR] | 144.5 [130, 172.5] | 139 [131, 176] | 140 [130, 173] |
| | Missing | 1 (4.0) | 2 (6.9) | 3 (5.6) |
| Blood pressure: diastolic (mmHg) | Median [IQR] | 80 [69.5, 86] | 79 [69, 90] | 80 [69, 87] |
| | Missing | 1 (4.0) | 2 (6.9) | 3 (5.6) |
| Cholesterol (mmol/L) | Median [IQR] | 4.6 [3.6, 5.5] | 4.4 [3.7, 5.5] | 4.5 [3.7, 5.5] |
| | Missing | 11 (44.0) | 12 (41.4) | 23 (42.6) |

BMI: Body Mass Index; CHD: Coronary Heart Disease; CKD: Chronic Kidney Disease; COPD: Chronic Obstructive Pulmonary Disease; IQR: Interquartile Range; mmHg: millimetres of mercury; mmol/L: millimoles Per Litre; N: number; PAD: Peripheral Artery Disease; SD: standard deviation; TIA: transient ischaemic attack

"[wife currently unwell]…*not feel able to commit the time and energy to completing the questionnaires and getting to a post office*" [TIA, declined].

**Randomisation.** All participants were randomised: 25 to the intervention arm and 29 to the control arm. The slight randomisation imbalance was due to a data entry error in minimisation category data for two participants.

Site staff were willing to randomise participants, but one nurse reflected on being disappointed for participants randomised to the control who she perceived could have benefitted from the intervention.

"*So you know sometimes you think you get the feeling that this person would really benefit from being spoken to again and they got the control I felt a bit gutted, but that's just the way it is, there's nothing you can do about that.*" [N2]

Intervention arm participants described being happy and excited to be in the intervention group, "*I was really pleased that I was going to get a follow up*" [P4, intervention]. Some control arm participants were disappointed with their allocation whereas others "*didn't mind*" [P9, control].

**Attrition.** Fig 2 shows the flow of participants through the trial. Initial recruitment rates were slower than anticipated; therefore, after consultation with the Study Oversight Committee, the 24-week follow-up was only completed by participants recruited up to 31st May 2022. As such, 70.4% (38/54) of the total sample were eligible to complete the 24-week questionnaire.

The return rate for the 1, 12, and 24-week questionnaires was 94.4% (51/54), 85.2% (46/54), and 71.1% (27/38), respectively (S2: Table S7). Response rates were influenced by reminder phone calls, emails and letters.

**Patient reported outcomes.** The patient reported outcomes were well completed with very little missing data (S2: Table S8).

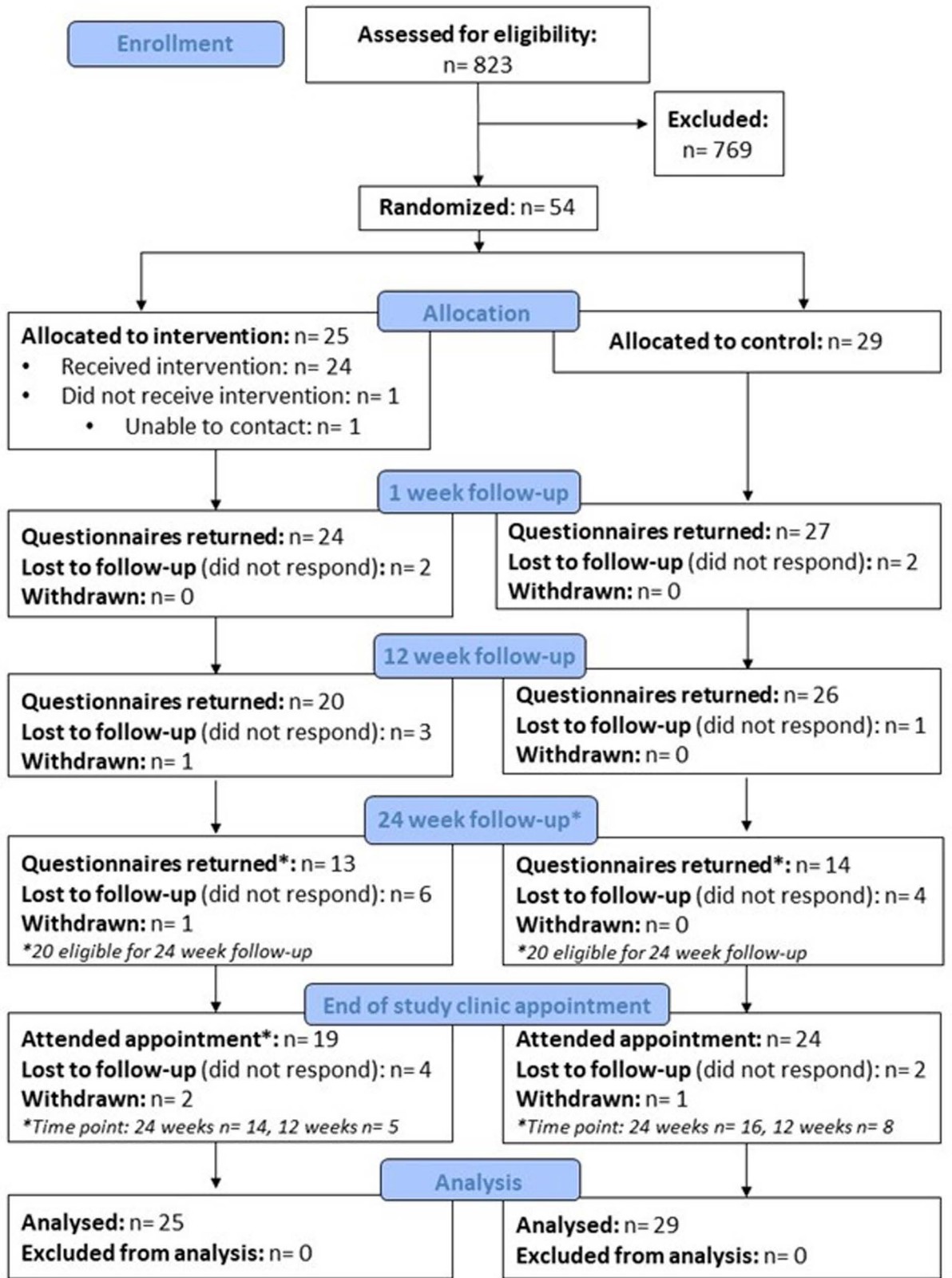

**Fig 2. Study flow diagram.**

Most participants reported the questionnaires were easy to complete (both paper and online) and did not find them too lengthy, *"Quick and easy because I was just able to tick questions"* [P9, control].

Some participants enjoyed completing questionnaires as it gave them an opportunity to reflect on how they were feeling.

*"But then there was questions about how you were feeling and they really did hit home… So it sort of stimulated questions in my own mind… I liked doing it because it was a time to actually think about how I was feeling."* [P10, control]

In contrast, a few participants found the questionnaires *"tiring… because I was so exhausted"* [P10, control], *"repetitive"* [P11, control] or *"laborious"* [P1, intervention], but were still willing to complete them. One participant had visual problems which affected completing the questionnaires.

Site staff reported that tracking and chasing overdue questionnaires was time consuming. Results of the patient reported outcomes and correlations are reported in Tables 4 and 5.

**Health economics.**  Overall, the health economics questionnaire was well completed (S2: Table S9). For the tables listing health care resource use and support services where a 'yes' or 'no' answer was requested, there were several participants who left this blank. Whilst this could be assumed to be equivalent to a 'no', the table could be reformatted to encourage better completion for a main trial.

Resources required to deliver the intervention were 30 minutes of staff time, estimated to be £50 for a band 6 and £61 for a band 7 nurse or allied health professional.

**End of study clinic appointment.**  Attendance rate for the end of study appointment was 79.6% (43/54), of which just under half were face-to-face appointments (46.5%; 20/43).

Qualitative analyses found most participants were happy to go back to the hospital for the end of study clinic appointment, particularly if it was arranged to coincide with an existing appointment. For some people having an additional cholesterol and blood pressure check was motivation to participate. A minority had transport issues or work commitments which prevented them attending.

*"Because what I tend to do is, when I ring them to book the end of study appointment I'll say, we're going to check your cholesterol, you last had it done at this point and it was that we'll re-check it and let you know in a couple of days, and then they're, okay. Because otherwise I don't know how routinely they would get it done at the GP, probably not very routinely."* [N2]

## Intervention: process evaluation

Findings are detailed below and summarised in Fig 3.

**Training.**  Training was delivered remotely by video call. Qualitative data demonstrated that, overall, site staff felt adequately prepared and knowledgeable about their roles in the feasibility study after the training, and there was consensus that the training was useful and covered essential aspects of the study.

*"I think yeah you definitely need the training to know like what we're actually wanting, what are people going to get out of it and what do we need to inform the GP of I suppose."* [N2]

**Structured follow-up appointment.**  Twenty-four out of 25 (96.0%) participants attended the intervention appointment. The intervention was delivered by four healthcare providers,

**Table 4. Patient reported outcome measure scores.**

| | | Post-baseline (n = 51) | 12 weeks (n = 46) | 24 weeks (n = 27) |
|---|---|---|---|---|
| **HADS** | | | | |
| HADS- Depression | **Normal (n, %)** | 34 (66.7) | 30 (65.2) | 20 (74.1) |
| | **Borderline (n, %)** | 12 (23.5) | 11 (23.9) | 5 (18.5) |
| | **Abnormal (n, %)** | 4 (7.8) | 3 (6.5) | 1 (3.7) |
| | **Missing (n, %)** | 1 (2.0) | 2 (4.4) | 1 (3.7) |
| | **Mean (SD)** | 5.1 (3.7) | 5.1 (3.8) | 4.6 (3.9) |
| HADS- Anxiety | **Normal (n, %)** | 24 (47.1) | 20 (43.5) | 17 (63.0) |
| | **Borderline (n, %)** | 14 (27.5) | 11 (23.9) | 4 (14.8) |
| | **Abnormal (n, %)** | 12 (23.5) | 11 (23.9) | 5 (18.5) |
| | **Missing (n, %)** | 1 (2.0) | 4 (8.7) | 1 (3.7) |
| | **Mean (SD)** | 7.2 (4.5) | 7.7 (4.7) | 6.7 (3.8) |
| Total score | **Mean (SD)** | 12.4 (7.3) | 12.8 (7.9) | 11.3 (6.9) |
| **EQ-5D-5L** | | | | |
| EQ-5D-5L index score | **Median [IQR]** | 0.87 [0.74, 0.92] | 0.85 [0.75, 0.92] | 0.85 [0.75, 0.92] |
| VAS | **Median [IQR]** | 70 [50, 90] | 75 [40, 90] | 77.5 [60, 90] |
| **PROMIS-10** | | | | |
| Physical Health | **Mean (SD)** | 45.5 (10.4) | 38.3 (5.6) | 39.4 (5.8) |
| Mental Health | **Mean (SD)** | 44.8 (9.4) | 41.1 (8.8) | 42.6 (6.8) |
| **FAS** | | | | |
| FAS Categories | **No fatigue (n, %)** | 25 (49.0) | 18 (39.1) | 13 (48.2) |
| | **Fatigue (n, %)** | 19 (37.3) | 20 (43.5) | 8 (29.6) |
| | **Extreme fatigue (n, %)** | 5 (9.8) | 6 (13.0) | 4 (14.8) |
| | **Missing (n, %)** | 1 (2.0) | 2 (4.4) | 2 (7.4) |
| Total score | **Mean (SD)** | 22.6 (8.4) | 23.5 (9.3) | 23.8 (9.3) |
| **MARS-5** | | | | |
| MARS-5 categories | **Low adherence (n, %)** | 25 (49.0) | 23 (50.0) | 13 (48.2) |
| | **Adherence (n, %)** | 25 (49.0) | 22 (47.8) | 12 (44.4) |
| | **Missing (n, %)** | 1 (2.0) | 1 (2.2) | 2 (7.4) |
| Total score | **Median [IQR]** | 24.5 [23,25] | 24 [23,25] | 24 [23,25] |
| **PAM-13** | | | | |
| Q1 (n, %) | **Disagree strongly** | 0 (0.0) | 1 (2.2) | 0 (0.0) |
| | **Disagree** | 1 (2.0) | 3 (6.5) | 0 (0.0) |
| | **Agree** | 24 (47.1) | 18 (39.1) | 14 (51.9) |
| | **Agree strongly** | 26 (51.0) | 22 (47.8) | 13 (48.2) |
| | **Not applicable** | 0 (0.0) | 0 (0.0) | 0 (0.0) |
| | **Missing** | 0 (0.0) | 2 (4.4) | 0 (0.0) |
| Q2 (n, %) | **Disagree strongly** | 0 (0.0) | 1 (2.2) | 0 (0.0) |
| | **Disagree** | 2 (3.9) | 1 (2.2) | 1 (3.7) |
| | **Agree** | 28 (54.9) | 23 (50.0) | 13 (48.2) |
| | **Agree strongly** | 20 (39.2) | 20 (43.5) | 13 (48.2) |
| | **Not applicable** | 1 (2.0) | 0 (0.0) | 0 (0.0) |
| | **Missing** | 0 (0.0) | 1 (2.2) | 0 (0.0) |
| Q3 (n, %) | **Disagree strongly** | 1 (2.0) | 2 (4.4) | 0 (0.0) |
| | **Disagree** | 3 (5.9) | 2 (4.4) | 5 (18.5) |
| | **Agree** | 30 (58.8) | 26 (56.5) | 12 (44.4) |
| | **Agree strongly** | 16 (31.4) | 15 (32.6) | 9 (33.3) |
| | **Not applicable** | 0 (0.0) | 0 (0.0) | 0 (0.0) |

*(Continued)*

**Table 4.** (Continued)

| | | Post-baseline (n = 51) | 12 weeks (n = 46) | 24 weeks (n = 27) |
|---|---|---|---|---|
| | Missing | 1 (2.0) | 1 (2.2) | 1 (3.7) |
| Q4 (n, %) | Disagree strongly | 1 (2.0) | 2 (4.4) | 0 (0.0) |
| | Disagree | 3 (5.9) | 2 (4.4) | 1 (3.7) |
| | Agree | 25 (49.0) | 22 (47.8) | 13 (48.2) |
| | Agree strongly | 21 (41.2) | 20 (43.5) | 13 (48.2) |
| | Not applicable | 1 (2.0) | 0 (0.0) | 0 (0.0) |
| | Missing | 0 (0.0) | 0 (0.0) | 0 (0.0) |
| Q5 (n, %) | Disagree strongly | 1 (2.0) | 0 (0.0) | 0 (0.0) |
| | Disagree | 2 (3.9) | 3 (6.5) | 2 (7.4) |
| | Agree | 32 (62.7) | 24 (54.2) | 13 (48.2) |
| | Agree strongly | 16 (31.4) | 19 (41.3) | 12 (44.4) |
| | Not applicable | 0 (0.0) | 0 (0.0) | 0 (0.0) |
| | Missing | 0 (0.0) | 0 (0.0) | 0 (0.0) |
| Q6 (n, %) | Disagree strongly | 0 (0.0) | 0 (0.0) | 0 (0.0) |
| | Disagree | 5 (9.8) | 5 (10.9) | 1 (3.7) |
| | Agree | 30 (58.8) | 23 (50.0) | 16 (59.3) |
| | Agree strongly | 15 (29.4) | 17 (37.0) | 9 (33.3) |
| | Not applicable | 1 (2.0) | 1 (2.2) | 0 (0.0) |
| | Missing | 0 (0.0) | 0 (0.0) | 1 (3.7) |
| Q7 (n, %) | Disagree strongly | 0 (0.0) | 0 (0.0) | 0 (0.0) |
| | Disagree | 2 (3.9) | 3 (6.5) | 1 (3.7) |
| | Agree | 27 (52.9) | 24 (54.2) | 15 (55.6) |
| | Agree strongly | 21 (41.2) | 19 (41.3) | 10 (37.0) |
| | Not applicable | 1 (2.0) | 0 (0.0) | 1 (3.7) |
| | Missing | 0 (0.0) | 0 (0.0) | 0 (0.0) |
| Q8 (n, %) | Disagree strongly | 0 (0.0) | 0 (0.0) | 0 (0.0) |
| | Disagree | 4 (7.8) | 3 (6.5) | 3 (11.1) |
| | Agree | 33 (64.7) | 24 (54.2) | 17 (63.0) |
| | Agree strongly | 12 (23.5) | 17 (37.0) | 6 (22.2) |
| | Not applicable | 1 (2.0) | 0 (0.0) | 1 (3.7) |
| | Missing | 1 (2.0) | 2 (4.4) | 0 (0.0) |
| Q9 (n, %) | Disagree strongly | 1 (2.0) | 0 (0.0) | 0 (0.0) |
| | Disagree | 7 (13.7) | 6 (13.1) | 5 (18.5) |
| | Agree | 34 (66.7) | 24 (54.2) | 16 (59.3) |
| | Agree strongly | 8 (15.7) | 14 (30.4) | 6 (22.2) |
| | Not applicable | 1 (2.0) | 1 (2.2) | 0 (0.0) |
| | Missing | 0 (0.0) | 1 (2.2) | 0 (0.0) |
| Q10 (n, %) | Disagree strongly | 0 (0.0) | 0 (0.0) | 0 (0.0) |
| | Disagree | 10 (19.6) | 8 (17.4) | 4 (14.8) |
| | Agree | 26 (51.0) | 22 (47.8) | 16 (59.3) |
| | Agree strongly | 15 (29.4) | 15 (32.6) | 7 (25.9) |
| | Not applicable | 0 (0.0) | 0 (0.0) | 0 (0.0) |
| | Missing | 0 (0.0) | 1 (2.2) | 0 (0.0) |
| Q11 (n, %) | Disagree strongly | 0 (0.0) | 0 (0.0) | 1 (3.7) |
| | Disagree | 7 (13.7) | 6 (13.0) | 8 (29.6) |
| | Agree | 33 (64.7) | 21 (45.7) | 13 (48.2) |

*(Continued)*

**Table 4.** (Continued)

| | | Post-baseline (n = 51) | 12 weeks (n = 46) | 24 weeks (n = 27) |
|---|---|---|---|---|
| | **Agree strongly** | 8 (15.7) | 16 (34.8) | 5 (18.5) |
| | **Not applicable** | 2 (3.9) | 1 (2.2) | 0 (0.0) |
| | **Missing** | 1 (2.0) | 2 (4.4) | 0 (0.0) |
| Q12 (n, %) | **Disagree strongly** | 0 (0.0) | 0 (0.0) | 0 (0.0) |
| | **Disagree** | 10 (19.6) | 8 (17.4) | 9 (33.3) |
| | **Agree** | 33 (64.7) | 20 (43.5) | 12 (44.4) |
| | **Agree strongly** | 5 (9.8) | 13 (28.3) | 6 (22.2) |
| | **Not applicable** | 2 (3.9) | 2 (4.4) | 0 (0.0) |
| | **Missing** | 1 (2.0) | 3 (6.5) | 0 (0.0) |
| Q13 (n, %) | **Disagree strongly** | 0 (0.0) | 1 (2.2) | 0 (0.0) |
| | **Disagree** | 11 (21.6) | 8 (17.4) | 6 (22.2) |
| | **Agree** | 34 (66.7) | 20 (43.5) | 16 (59.3) |
| | **Agree strongly** | 6 (11.8) | 14 (30.4) | 5 (18.5) |
| | **Not applicable** | 0 (0.0) | 1 (2.2) | 0 (0.0) |
| | **Missing** | 1 (2.0) | 2 (4.4) | 0 (0.0) |
| **Satisfaction with care (n, %)** | | | | |
| | **Very satisfied** | 29 (56.9) | 22 (47.8) | 10 (37.0) |
| | **Satisfied** | 10 (19.6) | 10 (21.7) | 7 (25.9) |
| | **Neutral** | 7 (13.7) | 11 (23.9) | 5 (18.5) |
| | **Dissatisfied** | 5 (9.8) | 1 (2.2) | 2 (7.4) |
| | **Very dissatisfied** | 0 (0.0) | 2 (4.4) | 1 (3.7) |
| | **Missing** | 0 (0.0) | 0 (0.0) | 2 (7.4) |

EQ-5D-5L: EuroQol 5-Dimensions; FAS: Fatigue Assessment Scale; HADS: Hospital Anxiety and Depression Scale; MARS: Medication Adherence Rating Scale -5; PAM-13: Patient Activation Measure-13; PROMIS-10: Patient-Reported Outcomes Measurement Information System - Global Health 10

all nurses. Most participants had one appointment, six participants had >1 appointment. Median appointment duration was 22.5 minutes (IQR 20, 30; range 10-65) (S2: Table S10). Participants were generally satisfied with appointment length and nurses were willing to provide additional follow-up appointments if required.

*"I: Yes, and how did you guys feel about having that potential option that people may be kind of calling you, how did that fit in with you clinical work?*

*IV: Oh I don't mind, I quite like that."* [N2]

The intervention appointment was intended to be face-to-face; however, COVID restrictions resulted in most appointments being conducted by telephone, only one was face-to-face. Qualitative data showed a positive response to telephone delivery from both participants and nurses. Telephone appointments were convenient, people felt more relaxed in their own home with less time pressure, and it was easier to discuss personal issues, such as sex. However, one participant who had hearing difficulties found telephone calls challenging.

*"Well I was quite happy with a phone call… And also it saves me having to keep going up and then the stress of trying to find somewhere to park at* [site 2] *as well, because that's always a nightmare. So it was nice because it was kind of in my own home and I was comfortable and relaxed instead of sitting outside with COVID and everything."* [P4]

*"I probably wouldn't have mentioned the sex thing, if it was face-to-face."* [P8]

**Table 5. Correlation of Patient reported outcome measures.**

| Postbaseline | | | | | |
| --- | --- | --- | --- | --- | --- |
| | EQ-5D-5L | HADS | FAS | PROMIS MH | PROMIS PH |
| EQ-5D-5L | 1 | | | | |
| HADS | -0.77 | 1 | | | |
| FAS | -0.67 | 0.71 | 1 | | |
| PROMIS MH | 0.67 | -0.85 | -0.74 | 1 | |
| PROMIS PH | 0.73 | -0.71 | -0.72 | 0.77 | 1 |
| 12 weeks | | | | | |
| | EQ-5D-5L | HADS | FAS | PROMIS MH | PROMIS PH |
| EQ-5D-5L | 1 | | | | |
| HADS | -0.71 | 1 | | | |
| FAS | -0.65 | 0.72 | 1 | | |
| PROMIS MH | -0.66 | 0.78 | 0.80 | 1 | |
| PROMIS PH | -0.56 | 0.66 | 0.78 | 0.73 | 1 |
| 24 weeks | | | | | |
| | EQ-5D-5L | HADS | FAS | PROMIS MH | PROMIS PH |
| EQ-5D-5L | 1 | | | | |
| HADS | -0.79 | 1 | | | |
| FAS | -0.64 | 0.66 | 1 | | |
| PROMIS MH | -0.63 | 0.84 | 0.67 | 1 | |
| PROMIS PH | -0.64 | 0.69 | 0.72 | 0.73 | 1 |

EQ-5D-5L: EuroQol 5-Dimensions; FAS: Fatigue Assessment Scale; HADS: Hospital Anxiety and Depression Scale; MARS: Medication Adherence Rating Scale -5; PAM-13: Patient Activation Measure-13; PROMIS-10: Patient-Reported Outcomes Measurement Information System - Global Health 10 (MH: Mental Health, PH: Physical Health)

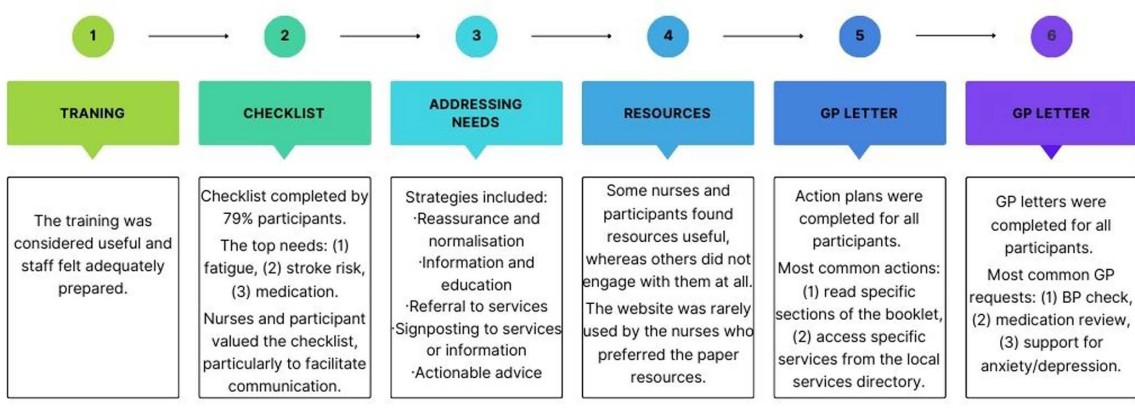

**Fig 3. summary of findings relating to the intervention process evaluation.**

No contamination with control was reported or observed. No adverse events were reported.

**Identifying needs: checklist.** Most (79.2%; 19/24) participants completed the checklist. The top three needs checked were fatigue (68.4%; 13/19), advice about stroke risk (63.2%; 12/19) and advice about stroke prevention medication (52.6%; 10/19) (S2: Table S11).

Nurses delivering the intervention valued the checklist to structure the appointment and facilitate communication.

**Table 6. Strategies to address needs evidenced by qualitative data, action plans, GP letters and intervention observations.**

| Data source | Quote |
|---|---|
| **Reassurance/ normalisation** | |
| **Qualitative data: participant** | *"Well, they said that it* [fatigue] *is quite normal. And I asked them how long it lasts, and they said, 'Well, it's difficult to say, because even with a TIA, some people are tired for six months, some people a year, and some people longer.' So reassurance that it was normal was helpful."* [P8] <br> *"I think it was important that someone is willing to listen to you and not treat you like you're off your head or 'Of course you have, don't be silly.' They were very respectful and very supportive. And you didn't feel it was rushed."* [P4] |
| **Qualitative data: nurse** | *"And a lot of it is just reassurance. Just giving them reassurance"* [N4] <br> *"[discussing a participant who was still off work] I think she wanted reassurance that yes this is normal, and you have to put your health first. I think she needed someone to say it's fine."* [N2] <br> *"[discussing supporting a participant with anxiety] …it's the reassurance and the opportunity to talk to someone. And do you know, it takes five minutes for a quick chat. But yeah, anxiety is probably the biggest."* [N4] <br> *"So I think when they were mentioning fatigue and I was saying, yes that's actually quite normal, I think people were relieved and they didn't actually realise, they thought they should have just got over it…"* [N2] |
| **Observation field notes** | *"Participant felt fatigue, but wasn't aware this could be due to their TIA, nurse reassured that this is common and normal."* [P12] |
| **GP letter** | *"[participants name] reported feelings of fatigue after the TIA, I have explained that this is very common"* [P15] |
| **Information/ education** | |
| **Qualitative data: participant** | **Medication:** *"I: So having that discussion with them helped you make the decision whether you should be taking the clopidogrel.* <br> *IV: Yes… It was reassuring, and it was beneficial to be able to talk to medical professionals in a non-rushed atmosphere."* [P8] <br> **Exercise:** *"I asked a lot of questions, I think, which they answered. And you do feel a bit alone…* [referring to generic stroke information] *'everything in moderation' is a bit broad, really, because my moderation might not be the person I'm speaking to's moderation. And it wasn't really specific enough… And going for a walk. Her moderate walk might be three miles. My moderate walk might be round the block."* [P8] <br> **Diagnosis:** *"They explained things in great detail and that and in a way that I could understand it without all these big medical words. They took the time to explain everything about it. It was brilliant… they took the time to explain it in layman's terms, which I could understand, and it's been brilliant. It has been really helpful."* [P5] |
| **Qualitative data: Nurse** | **Medication:** *"So it was educating them on the tablets. They're given medication really now, well at the time on the stroke unit, the new medication wasn't explained to them. So we would educate them on the importance of the medication, what it was doing, why it was being used."* [N4] <br> **Stroke symptoms:** *"I: So you found that people still weren't aware of what they needed to do if they had further symptoms?* <br> *IV: No, absolutely not, no. But that's a general problem. I say to all my patients, 'If you go home and develop symptoms what would you do?' They don't always say ring 999. They just don't. So it's just going about education."* [N4] <br> **Driving:** *"When can they drive, they weren't always given driving advice. Going back to work, going to the gym. It was the basics. But because of the constraints they're just not always given the advice at the time."* [N4] <br> **Imaging and investigations:** *"…we would give them whatever results had come back and then I would, if they hadn't had any results back I would keep an eye on it and I'd let them know once they'd got the results."* [N2] |

*(Continued)*

**Table 6.** (Continued)

| Data source | Quote |
|---|---|
| **Observation field notes** | **Medication:** "Participant had query about why they had been taken of one of their blood thinners – had previously been on two." [P12]<br>"Nurse checked understanding of medication" [P13]<br>**Alcohol:** "Advised to reduce alcohol intake – participant had not realised their alcohol intake was high: 2 beers and double whiskey a day" [P12]<br>**Imaging and investigations:** "Participant had not received imaging results back, nurse gave summary of results" [P12]<br>**Stroke risk:** "Participant had questions about future stroke risk – this was their main concern" [P13] |
| **GP Letter** | **Medication:** "[participant] currently takes Clopidogrel I have assured her she is on the correct medication and to continue with this." [P16] |
| **Direct referral to support services** | |
| **Qualitative data: participant** | **Positive:** "…*they sent me straight through to the stop smoking team and I've been getting patches and talking to Joanne every week. So they're helping me change my lifestyle.*" [P4]<br>**Negative:** "*I: … they said that they were going to get someone from a smoking clinic to call you, I think?*<br>*IV: Yeah. I've not had a call off them yet.*" [P5] |
| **Qualitative data: nurse** | "*Yes, a couple of people I've referred to smoking cessation, which I think otherwise they wouldn't have sort that out themselves. And the community smoking cessation team are really good they'll ring them that same day at home, otherwise even though I presume it must have been mentioned during the clinic appointment but the person hadn't thought, it obviously hadn't been mentioned that they could refer them, and it was only a simple like, we can refer them electronically, it literally takes you two minutes and they will ring them that same day and the balls rolling then.*" [N2] |
| **Signpost to services/ information** | |
| **Qualitative data: participant** | **Services:** "*She asked about groups or if there was anything I was interested in because there were different online groups and she gave me all that information. I had a lot of help out there really where I could have gone. I found it very, very helpful and useful.*" [P3]<br>**Information:** "*… like drink awareness and things like that, which explains how many units of things … So yeah, so it made me more aware of how much two cans is, so it helped me to work out where I was at that point and to where I'd like to be down to as in units of alcohol. It helped me understand all that.*" [P5] |
| **Qualitative data: nurse** | **Services:** "*…some people were really interested in joining a walking group and things like that…*" [N2] |
| **Action plan** | **Services:** "I have enclosed a directory of local sand have highlighted the services we spoke about in the appointment" [P18]<br>"I have highlighted the weight management services on page 2 this is a self-referral service for free weight loss support" [P20]<br>"Read the enclosed information on wellbeing and emotional support from the 'Local services directory' (Page 6) this is a self-referral service to help give you the tools to cope." [P19]<br>**Information:** "Access website [URL] which has lots of information about local groups" [P18]<br>"Read booklet included with this letter there are some resources identified to help with healthy eating and losing weight" [P20]<br>"Read the enclosed self help guide on anxiety" [P15] |
| **Observation field notes** | **Information:** "Participant felt anxious about having another stroke, nurse highlighted information and resources for anxiety in the booklet, will also inform GP." [P12] |

*(Continued)*

**Table 6.** (Continued)

| Data source | Quote |
|---|---|
| **GP Letter** | **Information:** "…she is very pro-active in improving her health and we have sent out further resources to help assist with these changes." [P17]<br>"[participant] reports feeling anxious following the TIA with regards to it reoccurring, she mentions she suffered anxiety prior to the episode, she does not feel like she needs any professional help to manage this but I have provided her with a self-help guide with tips and techniques to try and help manage this if needed. [P15]<br>"[participant] Mrs Lea reported feelings of fatigue after the TIA, I have explained that this is very common and I have provided resources from the Stroke Association to help [participant] manage this. [participant] also reports problems with memory and thinking and I have also provided some resources to try and assist with this." [P16] |
| **Actionable advice/ recommendations** | |
| **Qualitative data: participant** | *"I went in, they said, 'We advise you to get a monitor to monitor your blood pressure and that. We're going to give you some tablets to go bring your blood pressure down'. So only recently I've been doing my blood pressure monitoring and that since I had the high blood pressure and the TIA. They said they'd like to monitor them and then just write them down and when I see my GP, let him know where they're at, so yeah."* [P5] |
| **Action plan: Goal setting** | "She is going to set small goals and use the resources as a guide to help her. For instance: To be able to walk for 40 minutes again. She will try to achieve this by using the sit fit app and squeezy + sleepio. She will also cut down on alcohol before bed. She will try swimming again." [P21] |
| **Issues not related to TIA/ minor stroke** | |
| **Qualitative data: nurse** | *"… but then equally the things that weren't related to the TIA we would still discuss them because obviously it was still having an impact on everything else…"* [N2]<br>*"I think it's just about making the patient aware that you are just there to talk about the TIA and direct them back to the GP or if we know of any appropriate services that they can go back to, signposting them to them, I didn't have a problem with doing that."* [N3] |

*"That* [checklist] *was probably the best thing because they had something to sit and look at first and work through and I think that definitely lead the conversation… without that I think the appointment wouldn't have flowed as well…"* [N2]

Participants valued the checklist as a tool to prepare for and communicate during the appointment.

*"Yeah, straightforward and easy to understand and fill in … it prepared me for what kind of questions would be and things we'd be talking about, so yeah. It was really helpful."* [P5]

**Addressing needs.** Nurses used a range of strategies to address needs, which were tailored to participant needs and preferences, as demonstrated by qualitative data, action plans, General Practitioner (GP) letters and intervention observations (n = 3 observations; S2: Table S12). Strategies included: reassurance and normalisation; information and education; direct referral to support services; signposting to services or information; and actionable advice (Table 6). Occasionally participants raised issues unrelated to TIA or minor stroke, but nurses were happy to discuss these needs and signpost to support within reason.

Reassurance and normalisation Both participants and nurses highlighted the importance of reassurance and normalising symptoms or concerns, particularly regarding fatigue and fear

of another stroke. This was often what participants valued the most from the appointment and sometimes this was enough to address needs. Participants also valued the opportunity to be listened to. One participant who suffered from anxiety and fatigue had multiple follow-up phone calls where the nurse simply provided reassurance and information, which empowered the participant to make significant changes, such as changing to a less stressful job, "... *to be honest with you, I probably would have struggled and not got through as well if I'd have not had that basic lifeline. It was a lifeline to me*" [P3].

Information and education Participants valued the opportunity to ask questions, particularly about medication, stroke risk, lifestyle and future stroke symptoms. There were examples of this leading to participants changing their behaviour, such as deciding to take medication or reducing alcohol intake. Often participants had been given generic information and wanted personalised information or to clarify how to interpret advice, such as what does *"moderate"* mean. Participants commented that nurses used lay language and sometimes it was the first time their diagnosis, risk factors or medication had been explained to them in such terms. Nurses were also able to provide and explain results of investigations (e.g., imaging) that the participant had not received yet.

Direct referral to support services Two participants were referred to smoking cessation services. One participant was directly contacted and started accessing the service within a week; whereas, the other participant was not contacted by the service.

Signposting to services or information Nurses recommended specific services from the directory of local services, particularly for weight management, exercise groups and emotional support. They also highlighted specific sections of the self-management booklet, particularly for fatigue, anxiety and healthy eating, and other information resources, such as Stroke Association leaflets, websites or apps.

Actionable advice Some action plans included specific recommendations, advice or goal setting (such as walking for 40 minutes). Qualitative data provided examples of participants acting on advice, such as buying a blood pressure monitor and monitoring blood pressure at home.

**Resources.** On the intervention log, nurses self-reported that they used at least one of the provided resources in 87.5% (21/24) of appointments. Some nurses valued and used the resources.

*"… but I think the booklet was probably the most useful, because it was quite easy to read... If say, so particular fatigue, if they'd expressed right this is quite a problem, we'd direct them to the page …"* [N2]

The website was rarely used or promoted by the nurses who preferred the paper resources.

*"I don't think I mentioned the website, I think maybe I did on the first appointment, but I don't think I mentioned the website after that, we focused on the booklets and sending the booklets out"* [N3]

Similarly, some participants found the booklet and directory of services useful, particularly to address specific information needs, whereas others did not engage with them at all.

**Action plan.** An action plan was completed with all intervention participants. The most common actions for participants were to read specific sections of the TIA/minor stroke booklet (45.8%; 11/24), and access specific services from the directory of local services (41.7%; 10/24) (S2: Table S13). Five (20.8%) action plans had no specific actions included. The quality of the actions plans varied: some were detailed and addressed the participant personally;

whereas, others used acronyms, referred to the participant in third person and were brief. Some participants reported they found the action plan useful, whereas others did not recall seeing it or did not proactively use it.

**GP letter.** GP letters were completed for all intervention participants. The three most common GP requests were: blood pressure check (25.0%; 6/24), medication review (25.0%; 6/24) and support for anxiety, depression or low mood (25.0%; 6/24) (S2: Table S14). 41.7% (10/24) of GP letters had no specific actions for the GP.

Despite having a template for GP letters, the level of detail included varied. Similar to the action plans, some GP letters were well written and comprehensive, whereas others lacked detail. Qualitative data found that some nurses perceived that GPs would not engage with the GP letter, which made them less motivated to complete it.

*"I don't know whether the GP would actually action any of the things. If I'm honest I don't have that much confidence that they would read it and action things"* [N2]

**Acceptability.** Feedback questionnaires were completed by 16 participants. All participants reported they were very satisfied (14/16; 87.5%) or quite satisfied (2/16; 12.5%) with their appointment overall. All elements of the intervention – checklist, appointment and action plan – scored highly on the feedback questionnaire (S2: Table S15).

All interviewed intervention arm participants reported positive experiences. Often it was the only follow-up care they received. Similarly, nurses valued the intervention as providing support otherwise missing from current care.

*"So that I did feel helped and that was my support. It was second to none to be honest with you."* [P3]

*"… But she wouldn't have got any of that* [additional support] *if she hadn't have gone into the study. She'd have just been one of the numbers who was just discharged and that would've been it."* [N4]

Nurses delivering the intervention were described as respectful, caring, friendly and supportive.

*"The nurses doing it are wonderful, really caring and supportive and very friendly."* [P4]

Some people felt they did not have any needs and did not benefit from the appointment. Similarly, one nurses commented that not every single TIA or minor stroke patient would need the appointment, but it was difficult to identify who.

*"… it hasn't made any difference to me."* [P1]

*"I don't think every single person would need it but certainly a lot of people."* [N2]

## Discussion

### Principal findings

Nurse-led follow up after TIA and minor stroke, using a structured approach to identify and address needs, is feasible and acceptable. Overall, intervention fidelity was high and the intervention largely aligned with the logic model. The process evaluation illustrated how patients benefitted from the intervention through support not typically provided through usual care.

This included direct referral or signposting to support services, information and education, actionable advice, and reassurance about and normalisation of recovery.

The trial design was feasible and acceptable for both patients and clinical staff. Recruitment was lower than anticipated, probably because of the unanticipated impact of COVID on workload pressures and staff sickness.

## Interpretation of findings and comparison with literature

There are few interventions focused on care after TIA and minor stroke that address holistic needs. Scoping reviews demonstrate that previous research focused largely on secondary prevention and management of increased risk of further stroke.[27,28] Similar to our intervention, most studies delivered the intervention in an outpatient clinic setting and used low-tech and low-cost materials such as exercise booklets and diaries.

Our checklist was found to be useful in eliciting needs which might otherwise have been ignored and was valued by both patients and nurses. The checklist functioned as intended by providing an opportunity for participants to reflect on their needs prior to the appointment, facilitating communication, and structuring the appointment. These findings align with wider literature which demonstrates pre-consultation interventions, such a checklists, increase question asking and patient satisfaction.[29,30] However, there is a lack of evidence regarding whether checklists effectively result in patient benefit. Our checklist is an adapted version of the Stroke Review Checklist,[31] found to be practical for use in a primary care, perceived as useful by patients and healthcare providers, and resulted in agreed action plans. However, a definitive trial found no effects on emotional health or participation outcomes either individually or collectively as part of a complex intervention.[32]

In line with the logic model, nurses used a variety of strategies to address needs reflecting an individualised care approach to the diversity of needs and participants' preferences. In addition to referral and signposting, participants and nurses identified the importance of reassurance and normalisation, particularly to address fatigue and anxiety. Although the intervention was originally planned to be delivered face-to-face, telephone delivery was acceptable and convenient for participants. Telehealth is increasingly being adopted in post-pandemic stroke care with patients and healthcare providers reporting satisfaction with remote delivery;[33] however, evidence of effectiveness is lacking.[34]

There was a mixed response to the resources with some nurses and participants valuing them whereas others did not engage, but it was unclear why. Similarly, all participants had an action plan, but there was a mixed response to how participants engaged with these. This may reflect diversity in quality of the action plans and relevance to participant needs. It is important to acknowledge that some participants felt they did not have any needs and did not benefit from the appointment. However, it is difficult to identify which patients need follow-up, particularly given the diversity of needs which encompass information, secondary prevention and holistic needs.

It was unclear if the GP letter served as a tool to communicate with primary care or whether GPs actioned recommendations. Qualitative data suggests the level of detail nurses included in the letter was influenced by their perception of whether GPs would engage with the letter. Generalist and specialist silo working, fragmented care and lack of communication between primary and secondary care providers has previously been reported in the context of stroke and TIA care.[35–38] Improving communication and integrated care across healthcare settings is critical, but challenging. Barriers include: different information technology systems, time constraints, different technical languages, lack of specific information/ instructions, and perception of roles and responsibilities.[32,35,37] A UK intervention designed to enhance

communication pathways between specialist stroke care and primary care proved to be logistically difficult and was substituted with one-way video messages.[32]

## Strengths and limitations

Strengths of our study include: multi-site recruitment, robust study design, and mixed-methods process evaluation which explored feasibility and acceptability of both the intervention and trial design. However, a key limitation is that we were unable to capture data on whether GPs received or engaged with the intervention's GP letter. Therefore, we do not know if this element of the intervention functioned as intended to improve communication between secondary and primary care as theorised in our logic model. In terms of process evaluation data collection, all three intervention observations were from the same site, due to time and resource constraints. Another important limitation is that all participants were of white ethnicity, which may limit the generalisability of our findings.

## Implications for clinical practice and policy

Although we are yet to establish effectiveness and cost-effectiveness of our intervention, this feasibility study demonstrates the potential benefits of our intervention to identify and address variable needs after TIA and minor stroke. In particular, the checklist could be used in current care as a relatively low-cost way to improve communication and identification of needs. Importantly, our previous research demonstrated stroke clinicians often lack of knowledge of needs after TIA and minor stroke and are not aware of the lack of support from primary care. [7,37] Therefore, the training element of our intervention should not be overlooked.

There is variation in terms of TIA clinic care pathways in the UK.(7) Therefore, implementation requirements for our follow-up pathway would be context specific. For example, where a nurse-led follow-up is already in place, implementation may only require restructuring to incorporate intervention components such as the checklist and resources. However, for services with no nurse follow-up, implementation would require commissioning. Further research is required to explore barriers and enablers to implementation in different TIA clinic settings.

National guidelines currently focus on secondary prevention for TIA and minor stroke follow-up care; however, these could be updated to highlight holistic needs, such as fatigue and anxiety. People's needs after TIA and minor stroke vary; therefore, guidelines should promote proactive identification of needs, facilitated by a checklist, which encompass information provision, stroke prevention and holistic care. In terms of addressing needs, guidelines should emphasise the need for individualised care and importance of reassurance and normalisation.

## Implications for future research

Our progression criteria were met and indicate that a definitive evaluation of effectiveness and cost-effectiveness of the intervention is feasible.

As expected, we found heterogeneity in intervention participant needs and subsequent care. However, this presents a challenge for selecting a primary outcome measure for a trial to capture the diverse potential patient benefit reported from qualitative data. Secondary outcomes should be included to capture "softer" benefits, such as reassurance and normalisation, which are meaningful to patients.

To increase the diversity of the participant sample, future research should use frameworks to facilitate representative and equitable sample selection,[39,40] co-design research with under-served groups (such as recruitment strategies),[41] address known barriers,[42] and purposively select hospital sites from regions with diverse populations.[43] Future research to

improve/ facilitate communication between secondary and primary care would be valuable to further enhance the intervention.

## Conclusion

Nurse-led, structured, telephone follow up after TIA and minor stroke is feasible, acceptable and valued by patients and clinical staff. Healthcare providers should consider incorporating a checklist, which encompasses information provision, stroke prevention and holistic care, to improve communication and identification of needs. Future research should move to a full-scale definitive trial and health economics evaluation to evaluate the effectiveness and cost-effectiveness of the intervention.

## Supporting information

**S1 Checklist. CONSORT checklist.**
(DOCX)

**S2 File. Supporting tables and figures.**
(DOCX)

## Acknowledgments

We are grateful to all the participants who consented to participate in SUPPORT-TIA and thank all the hospital site staff who were involved in the set up and delivery of the feasibility study: Kirubananthan Nagaratnam, Neelima Bhupathiraju, Julie Didcock, Riturani Enjep-urapu, Kinza Emmanuel, Sarah Liderth, Habib Rehman, Pamela Farren, Cheryl Finch, Josh Copper, Glyn Fletcher and Paula Lopez. We thank all the patient partners who supported the entirety of the research from grant application to dissemination. The study was supported by the Birmingham Clinical Trials Unit (BCTU) who were instrumental in the protocol development and delivery of the trial: Khaled Ahmed, Marie Chadburn, Ryan Griffin, Sarah Tearne (Trials Management), Kelly Handley, Smitaa Patel (Statistics), James Brown, Gurmail Rai (Programming). We thank the Study Oversight Committee: Philip Collis, Jemma Hawkins, Timothy Pickles, James Sheppard (chair).

## Author contributions

**Conceptualization:** Grace M Turner, Melanie Calvert, Robbie Foy, Lou Atkins, Sarah Tearne, Jonathan Mant.

**Data curation:** Grace M Turner.

**Formal analysis:** Grace M Turner.

**Funding acquisition:** Grace M Turner.

**Methodology:** Grace M Turner, Melanie Calvert, Robbie Foy, Lou Atkins, Sarah Tearne, Sue Jowett, Kelly Handley, Philip Collis, Jonathan Mant.

**Project administration:** Grace M Turner.

**Supervision:** Melanie Calvert, Robbie Foy, Lou Atkins, Sarah Tearne, Sue Jowett, Kelly Handley, Philip Collis, Jonathan Mant.

**Writing – original draft:** Grace M Turner.

**Writing – review & editing:** Melanie Calvert, Robbie Foy, Lou Atkins, Sarah Tearne, Sue Jowett, Kelly Handley, Philip Collis, Jonathan Mant.

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
