## [Decision Letter · Decision Letter 0]

11 Nov 2024

PONE-D-24-33701Structured follow-up pathway to address unmet needs after transient ischaemic attack and minor stroke (SUPPORT TIA): feasibility study and process evaluationPLOS ONE

Dear Dr. Turner,

Thank you for submitting your manuscript to PLOS ONE. After careful consideration, we feel that it has merit but does not fully meet PLOS ONE’s publication criteria as it currently stands. Therefore, we invite you to submit a revised version of the manuscript that addresses the points raised during the review process.

**ACADEMIC EDITOR: ** After a critical external peer review by two experts, I recommended a minor revision to improve the paper's clarity and presentation based on the reviewers' concerns. Please see the attached reviewer comments below.

We look forward to receiving your revised manuscript.

Kind regards,

Dr Redoy Ranjan, MBBS, MRCSEd, Ch.M., MS (CV&TS), FACS

Academic Editor

PLOS ONE

Journal Requirements:

3. Please note that the grant information you provided in the ‘Funding Information’ and ‘Financial Disclosure’ sections do not match.

[GT received grant funding to deliver this project from a National Institute for Health and Care Research (NIHR) Post-Doctoral Fellowship Scheme grant number PDF-2017-10-047. RF receives grant funding from the NIHR (with funds paid to his institution) and chairs the NICE Implementation Strategy Group (non-paid). SJ reports grants from NIHR and Wellcome outside the submitted work. MC receives grant funding from the NIHR Birmingham Biomedical Research Centre, Health Data Research UK, Innovate UK, Macmillan Cancer Support, GSK, UCB Pharma, Research England, European Commission and EFPIA, Brain Tumor Charity, Gilead, Janssen, NIHR, UKRI, UK Research and Innovation, Merck; Royalties or licenses from Symptom Burden Questionnaire-Long COVID (as part of development team received revenue share from commercial license); Consulting fees from Aparito Ltd, CIS Oncology, Takeda, Merck, Daiichi Sankyo, Glaukos, GSK, PCORI, Genentech, Vertex, ICON, Halfloop, Pfizer; Payment or honoraria for lecture fees from University of Maastricht, reviewer fees from South-Eastern Norway Regional Health Authority and Singapore National Medical Research Council, speaker fee from Cochrane Portugal. PC, JM, KH and LA declare no conflict of interest.].

5. Please upload a copy of Figure 3, to which you refer in your text on page 16. If the figure is no longer to be included as part of the submission please remove all reference to it within the text.

Reviewers' comments:

Reviewer's Responses to Questions

**Comments to the Author**

1. Is the manuscript technically sound, and do the data support the conclusions?

Reviewer #1: Yes

Reviewer #2: Yes

2. Has the statistical analysis been performed appropriately and rigorously? 

Reviewer #1: Yes

Reviewer #2: N/A

3. Have the authors made all data underlying the findings in their manuscript fully available?

Reviewer #1: Yes

Reviewer #2: Yes

4. Is the manuscript presented in an intelligible fashion and written in standard English?

Reviewer #1: Yes

Reviewer #2: Yes

5. Review Comments to the Author

Reviewer #1: Dear Authors,

I have completed my review of your manuscript, "Structured follow-up pathway to address unmet needs after transient ischemic attack and minor stroke (SUPPORT TIA): feasibility study and process evaluation".I commend the effort put into this important study, which addresses a critical aspect of post-stroke care.

Overall Feedback: Your research is timely and could significantly improve patient outcomes. However, I believe the manuscript could benefit from further clarification and enhancement in several areas:

Suggestions for Improvement:

1.Introduction:

Expand on the background to better highlight unmet needs in TIA and minor stroke follow-up care.

Clearly state the research gap and how your study addresses it.

2.Methodology:

Provide more details on participant selection criteria, including any exclusion criteria and how the sample size was determined.

Clarify the specific qualitative and quantitative methods used in the process evaluation and any validation techniques.

3.Results:

Offer a more detailed interpretation of the data, discussing potential confounding factors.

Consider adding figures to visually represent key data points.

4.Discussion:

Strengthen the connection between your findings and existing literature. Highlight how your results compare with or diverge from previous studies.

Emphasize the clinical implications of your findings and how the follow-up pathway could be implemented in practice.

5.Conclusion:

Clearly outline the next steps for research and offer concrete recommendations for healthcare providers based on your findings.

Minor Revisions:

Ensure all abbreviations are defined upon first use.

Improve the readability of the manuscript with thorough proofreading.

Ethical Standards Compliance: I noted that ethical approval was obtained from the relevant IRB, and informed consent was collected from all participants. This adherence to ethical standards is crucial and ensures respect for participants' rights.

Adherence to Reporting Guidelines: The manuscript follows relevant reporting guidelines, such as CONSORT, which enhances its clarity and transparency. The detailed account of study design, methodology, and results aligns with the expected standards for research reproducibility.

Thank you for the opportunity to review your valuable work. I believe these changes will strengthen your manuscript and increase its chances of publication. Please feel free to reach out if you need further clarification on my comments.

Thank you for the opportunity to review this valuable work.

Reviewer #2: This manuscript is excellently written with a very clear purpose and aim. It is well structured. The figures are legible without too much detail in boxes. The message is an important one that can help with holistic provision of care to patients following TIA. The acronym is great SUPPORT TIA. No major concerns or suggestions from me.

6. PLOS authors have the option to publish the peer review history of their article (what does this mean? ). If published, this will include your full peer review and any attached files.

**Do you want your identity to be public for this peer review?** For information about this choice, including consent withdrawal, please see our Privacy Policy .

Reviewer #1: **Yes: ** Akram Aldilaimi

Reviewer #2: **Yes: ** Samyami S Chowdhury

---

## [Author Response · Author response to Decision Letter 1]

19 Dec 2024

Thank you for the opportunity to revise our manuscript. We have submitted a rebuttal letter that responds to each point raised by the reviewers.

---

## [Decision Letter · Decision Letter 1]

30 Dec 2024

Structured follow-up pathway to address unmet needs after transient ischaemic attack and minor stroke (SUPPORT TIA): feasibility study and process evaluation

PONE-D-24-33701R1

Dear Dr. Turner,

We’re pleased to inform you that your manuscript has been judged scientifically suitable for publication and will be formally accepted for publication once it meets all outstanding technical requirements.

Kind regards,

Dr Redoy Ranjan, MBBS, MRCSEd, Ch.M., MS (CV&TS), FACS

Academic Editor

PLOS ONE

**Comments to the Author**

**Review Comments to the Author**

Reviewer #1: (No Response)

---

## [Editor Report · Acceptance letter]

PONE-D-24-33701R1

PLOS ONE

Dear Dr. Turner,

I'm pleased to inform you that your manuscript has been deemed suitable for publication in PLOS ONE. Congratulations! Your manuscript is now being handed over to our production team.

Kind regards,

on behalf of

Dr. Redoy Ranjan

Academic Editor

PLOS ONE